# EXPERIENCE REPLAY FOR CONTINUAL LEARNING

## ABSTRACT

Continual learning is the problem of learning new tasks or knowledge while protecting old knowledge and ideally generalizing from old experience to learn new tasks faster. Neural networks trained by stochastic gradient descent often degrade on old tasks when trained successively on new tasks with different data distributions. This phenomenon, referred to as catastrophic forgetting, is considered a major hurdle to learning with non-stationary data or sequences of new tasks, and prevents networks from continually accumulating knowledge and skills. We examine this issue in the context of reinforcement learning, in a setting where an agent is exposed to tasks in a sequence. Unlike most other work, we do not provide an explicit indication to the model of task boundaries, which is the most general circumstance for a learning agent exposed to continuous experience. While various methods to counteract catastrophic forgetting have recently been proposed, we explore a straightforward, general, and seemingly overlooked solution – that of using experience replay buffers for all past events – with a mixture of on- and off-policy learning, leveraging behavioral cloning. We show that this strategy can still learn new tasks quickly yet can substantially reduce catastrophic forgetting in both Atari and DMLab domains, even matching the performance of methods that require task identities. When buffer storage is constrained, we confirm that a simple mechanism for randomly discarding data allows a limited size buffer to perform almost as well as an unbounded one.

## 1 INTRODUCTION

Modern day reinforcement learning (RL) has benefited substantially from a massive influx of computational resources. In some instances, the number of data points to feed into RL algorithms has kept in step with computational feasibility. For example, in simulation environments or in self-play RL, it is possible to generate fresh data on the fly. In such settings, the continual learning problem (Ring, 1997) is often ignored because new experiences can be collected on demand, and the start states of the simulation can be controlled. When training on multiple tasks, it is possible to train on all environments simultaneously within the same data batch.

As RL is increasingly applied to problems in industry or other real-world settings, however, it is necessary to consider cases, such as robotics, where gathering new experience is expensive or difficult. In such examples, simultaneous training may be infeasible. Instead, an agent must be able to learn from only one task at a time. The time spent on different tasks and the sequence in which those tasks occur are not under the control of the agent. The boundaries between tasks, in fact, will often be unknown – or tasks will deform continuously and not have definite boundaries at all. Such a paradigm for training eliminates the possibility of simultaneously acting upon and learning from several tasks, and leads to the danger of *catastrophic forgetting*, wherein an agent forgets what it has learned previously when it encounters a new situation.

Here, we consider the setting of reinforcement learning where compute and memory resources are large, but the environment is not stationary: this may arise because an RL agent is encountering a task curriculum or sequence of unrelated tasks, engaged in a budgeted physical interaction within a robot, or learning from unstructured interaction with humans. In this setting, the problem of continual learning rears its head: the distribution over experiences is not controlled to facilitate the agent's maintenance of previously acquired ability.

An ideal continual learning system should meet three requirements. First, it should retain previously learned capacities. When a previously encountered task or situation is encountered, performance

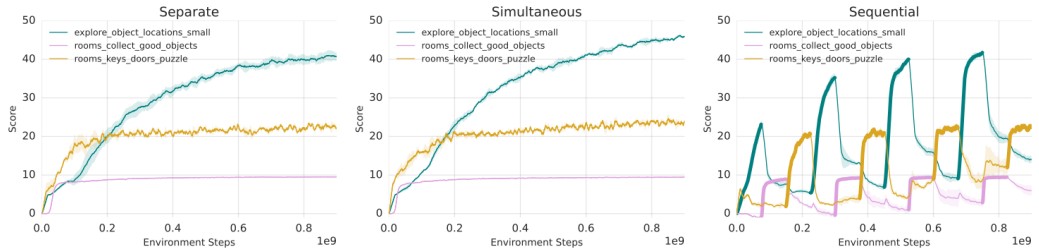

Figure 1: Separate, simultaneous, and sequential training: the $x$-axis denotes environment steps summed across all tasks and the $y$-axis episode score. In "Sequential", thick line segments are used to denote the task currently being trained, while thin segments are plotted by evaluating performance without learning. In simultaneous training, performance on `explore_object_locations_small` is higher than in separate training, an example of modest constructive interference. In sequential training, tasks that are not currently being learned exhibit very dramatic catastrophic forgetting. (See Appendix C for a different plot of these data.)

should immediately be good – ideally as good as it was historically. Second, maintenance of old skills or knowledge should not inhibit further rapid acquisition of a new skill or knowledge. These two simultaneous constraints – maintaining the old while still adapting to the new – represent the challenge known as the *stability-plasticity* dilemma Grossberg (1982). Third, where possible, a continual learning system should learn new skills that are related to old ones faster than it would have *de novo*, a property known as *constructive interference* or *positive transfer*.

We here demonstrate the surprising power of a simple approach: Continual Learning with Experience And Replay (CLEAR). We show that training a network on a mixture of novel experience on-policy and replay experience off-policy allows for both maintenance of performance on earlier tasks and fast adaptation to new tasks. A significant further boost in performance and reduction in catastrophic forgetting is obtained by enforcing behavioral cloning between the current policy and its past self. While memory is rarely severely limited in modern RL, we show that small replay buffers filled with uniform samples from past experiences can be almost as effective as buffers of unbounded size. When comparing CLEAR against state-of-the-art approaches for reducing catastrophic forgetting, we obtain better or comparable results, despite the relative simplicity of our approach; yet, crucially, CLEAR requires no information about the identity of tasks or boundaries between them.

## 2 RELATED WORK

The problem of catastrophic forgetting in neural networks has long been recognized (Grossberg, 1982), and it is known that rehearsing past data can be a satisfactory antidote for some purposes (McClelland, 1998; French, 1999). Consequently, in the supervised setting that is the most common paradigm in machine learning, catastrophic forgetting has been accorded less attention than in cognitive science or neuroscience, since a fixed dataset can be reordered and replayed as necessary to ensure high performance on all samples.

In recent years, however, there has been renewed interest in overcoming catastrophic forgetting in RL contexts and in supervised learning from streaming data (Parisi et al., 2018). Current strategies for mitigating catastrophic forgetting have primarily focused on schemes for protecting the parameters inferred in one task while training on another. For example, in Elastic Weight Consolidation (EWC) (Kirkpatrick et al., 2017), weights important for past tasks are constrained to change more slowly while learning new tasks. The Progressive Networks approach (Rusu et al., 2016) freezes subnetworks trained on individual tasks, and Progress & Compress (Schwarz et al., 2018) uses EWC to consolidate the network after each task has been learned. Kaplanis et al. (2018) treat individual synaptic weights as dynamical systems with latent dimensions / states that protect information. Outside of RL, Zenke et al. (2017) develop a method similar to EWC that maintains estimates of the importance of weights for past tasks, Li & Hoiem (2017) leverage a mixture of task-specific and shared parameters, and Milan et al. (2016) develop a rigorous Bayesian approach for estimating unknown task boundaries. Notably all these methods assume that task identities or boundaries are

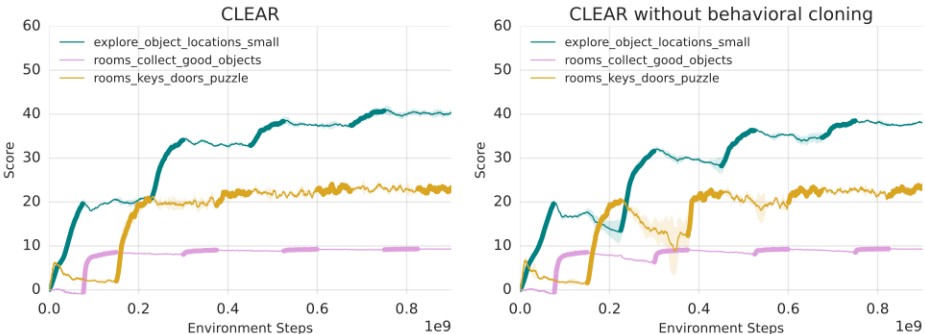

Figure 2: Demonstration of CLEAR on three DMLab tasks, which are trained cyclically in sequence. CLEAR reduces catastrophic forgetting so significantly that sequential tasks train almost as well as simultaneous tasks (see Figure 1). When the behavioral cloning loss terms are ablated, there is still reduced forgetting from off-policy replay alone. As above, thicker line segments are used to denote the task that is currently being trained. (See Appendix C for a different plot of these data.)

known, with the exception of Milan et al. (2016), for which the approach is likely not scalable to highly complex tasks.

Rehearsing old data via experience replay buffers is a common technique in RL. However, their introduction has primarily been driven by the goal of data-efficient learning on single tasks (Lin, 1992; Mnih et al., 2015; Gu et al., 2017). Research in this vein has included prioritized replay for maximizing the impact of rare experiences (Schaul et al., 2016), learning from human demonstration data seeded into a buffer (Hester et al., 2017), and methods for approximating replay buffers with generative models (Shin et al., 2017). A noteworthy use of experience replay buffers to protect against catastrophic forgetting was demonstrated in Isele & Cosgun (2018) on toy tasks, with a focus on how buffers can be made smaller. Previous works (Gu et al., 2017; O'Donoghue et al., 2016; Wang et al., 2016) have explored mixing on- and off-policy updates in RL, though these were focused on speed and stability in individual tasks and did not examine continual learning. Here, in CLEAR, we demonstrate that a mixture of replay data and fresh experience protects against catastrophic forgetting while also permitting fast learning, and performs better than either pure on-policy learning or pure off-policy learning from replay. We provide a thorough investigation and robust algorithm in CLEAR, conducting a wide variety of tests on both limited-size and unbounded buffers with complex RL tasks using state-of-the-art methods, and improve the stability of simple replay with the addition of behavioral cloning.

## 3 THE CLEAR METHOD

CLEAR uses actor-critic training on a mixture of new and replayed experiences. In the case of replay experiences, two additional loss terms are added to induce behavioral cloning between the network and its past self. The motivation for behavioral cloning is to prevent network output on replayed tasks from drifting while learning new tasks. We penalize (1) the KL divergence between the historical policy distribution and the present policy distribution, (2) the L2 norm of the difference between the historical and present value functions. Formally, this corresponds to adding the following loss functions, defined with respect to network parameters $\theta$:

$$L_{\text{policy-cloning}} := \sum_a \mu(a|h_s) \log \frac{\mu(a|h_s)}{\pi_\theta(a|h_s)},$$

$$L_{\text{value-cloning}} := ||V_\theta(h_s) - V_{\text{replay}}(h_s)||_2^2,$$

where $\pi_\theta$ denotes the (current) policy of the network over actions $a$, $\mu$ the policy generating the observed experience, and $h_s$ the hidden state of the network at time $s$. Note that computing $\text{KL}[\mu||\pi_\theta]$ instead of $\text{KL}[\pi_\theta||\mu]$ ensures that $\pi_\theta(a|h_s)$ is nonzero wherever the historical policy is as well.

We apply CLEAR in a distributed training context based on the Importance Weighted Actor-Learner Architecture (Espeholt et al., 2018). A single learning network is fed experiences (both novel and

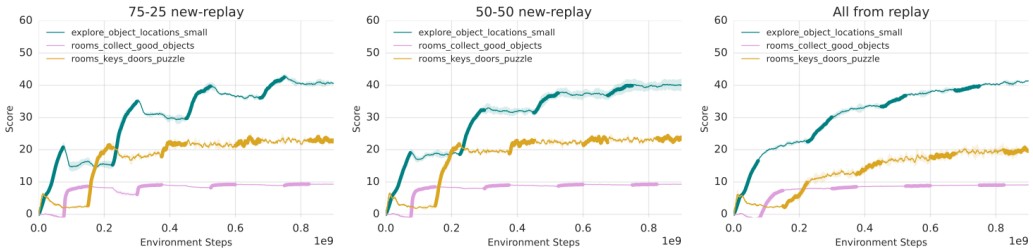

Figure 3: Comparison between different proportions of new and replay examples while cycling training among three tasks. We observe that with a 75-25 new-replay split, CLEAR eliminates most, but not all, catastrophic forgetting. At the opposite extreme, 100% replay prevents forgetting at the expense of reduced overall performance (with an especially noticeable reduction in early performance on the task `rooms_keys_doors_puzzle`). A 50-50 split represents a good tradeoff between these extremes. (See Appendix C for a different plot of these data.)

replay) by a number of acting networks, for which the weights are asynchronously updated to match those of the learner. The network architecture and hyperparameters are chosen as in Espeholt et al. (2018). Training proceeds according to V-Trace. Namely, define the V-Trace target $v_s$ by:

$$v_s := V(h_s) + \sum_{t=s}^{s+n-1} \gamma^{t-s} \left( \prod_{i=s}^{t-1} c_i \right) \delta_t V,$$

where $\delta_t V := \rho_t \left( r_t + \gamma V(h_{t+1}) - V(h_t) \right)$, $c_i := \min(\bar{c}, \frac{\pi_\theta(a_i|h_i)}{\mu(a_i|h_i)})$, and $\rho_t = \min(\bar{\rho}, \frac{\pi_\theta(a_t|h_t)}{\mu(a_t|h_t)})$, for constants $\bar{c}$ and $\bar{\rho}$.

Then, the value function update is given by the L2 loss:

$$L_{\text{value}} := \left( V_\theta(h_s) - v_s \right)^2.$$

The policy gradient loss is:

$$L_{\text{policy-gradient}} := -\rho_s \log \pi_\theta(a_s|h_s) \left( r_s + \gamma v_{s+1} - V_\theta(h_s) \right).$$

We also use an entropy loss:

$$L_{\text{entropy}} := \sum_a \pi_\theta(a|h_s) \log \pi_\theta(a|h_s).$$

The loss functions $L_{\text{value}}$, $L_{\text{policy-gradient}}$, and $L_{\text{entropy}}$ are applied both for new and replay experiences. In addition, we add $L_{\text{policy-cloning}}$ and $L_{\text{value-cloning}}$ for replay experiences only. In general, our experiments use a 50-50 mixture of novel and replay experiences, though performance does not appear to be very sensitive to this ratio. Further implementation details are given in Appendix A.

## 4 RESULTS

### 4.1 CATASTROPHIC FORGETTING VS. INTERFERENCE

Our first experiment (Figure 1) was designed to distinguish between two distinct concepts that are sometimes conflated, *interference* and *catastrophic forgetting*, and to emphasize the outsized role of the latter as compared to the former. Interference occurs when two or more tasks are incompatible (*destructive interference*) or mutually helpful (*constructive interference*) within the same model. Catastrophic forgetting occurs when a task's performance goes down not as a result of incompatibility with another task but as a result of the second task overwriting it within the model. As we aim to illustrate, the two are independent phenomena, and while interference may happen, forgetting is ubiquitous.

We considered a set of three distinct tasks within the DMLab set of environments (Beattie et al., 2016), and compared three training paradigms on which a network may be trained to perform these

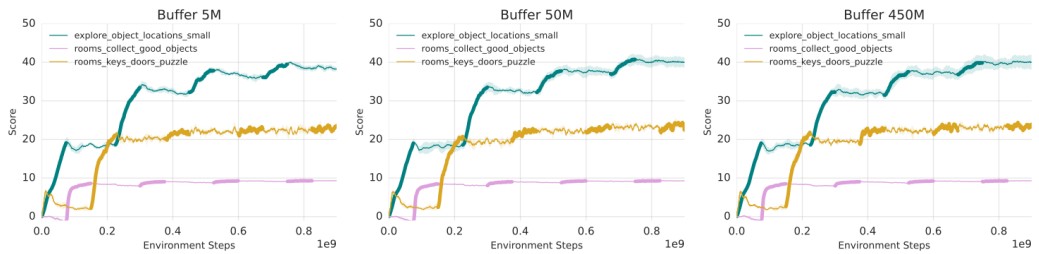

Figure 4: Comparison of the performance of training with different buffer sizes, using reservoir sampling to ensure that each buffer stores a uniform sample from all past experience. We observe only minimal difference in performance, with the smallest buffer (storing 1 in 200 experiences) demonstrating some catastrophic forgetting. (See Appendix C for a different plot of these data.)

three tasks: (1) Training networks on the individual tasks *separately*, (2) training a single network examples from all tasks *simultaneously* (which permits interference among tasks), and (3) training a single network *sequentially* on examples from one task, then the next task, and so on cyclically. Across all training protocols, the total amount of experience for each task was held constant. Thus, for separate networks training on separate tasks, the $x$-axis in our plots shows the total number of environment frames summed across all tasks. For example, at three million frames, one million were on task 1, one million on task 2, and one million on task 3. This allows a direct comparison to simultaneous training, in which the same network was trained on all three tasks.

We observe that in DMLab, there is very little difference between separate and simultaneous training. This indicates minimal interference between tasks. If anything, there is a small amount of constructive interference, with simultaneous training performing slightly better than separate training. We assume this is a result of (i) commonalities in image processing required across different tasks, and (ii) certain basic exploratory behaviors, e.g., moving around, that are advantageous across tasks. (By contrast, destructive interference might result from incompatible behaviors or from insufficient model capacity.)

By contrast, there is a large difference between either of the above modes of training and sequential training, where performance on a task decays immediately when training switches to another task – that is, catastrophic forgetting. Note that the performance of the sequential training appears at some points to be greater than that of separate training. This is purely because in sequential training, training proceeds exclusively on a single task, then exclusively on another task. For example, the first task quickly increases in performance since the network is effectively seeing three times as much data on that task as the networks training on separate or simultaneous tasks.

## 4.2 CLEAR

We here demonstrate the efficacy of CLEAR for diminishing catastrophic forgetting (Figure 2). We apply CLEAR to the cyclically repeating sequence of DMLab tasks used in the preceding experiment. Our method effectively eliminates forgetting on all three tasks, while preserving overall training performance (see "Sequential" training in Figure 1 for reference). When the task switches, there is little, if any, dropoff in performance when using CLEAR, and the network picks up immediately where it left off once a task returns later in training. Without behavioral cloning, the mixture of new experience and replay still reduces catastrophic forgetting, though the effect is reduced.

### 4.3 BALANCE OF ON- AND OFF-POLICY LEARNING

In this experiment (Figure 3), we consider the ratio of new examples to replay examples during training. Using 100% new examples is simply standard training, which as we have seen is subject to dramatic catastrophic forgetting. At 75-25 new-replay, there is already significant resistance to forgetting. At the opposite extreme, 100% replay examples is extremely resistant to catastrophic forgetting, but at the expense of a (slight) decrease in performance attained. We believe that 50-50 new-replay represents a good tradeoff, combining significantly reduced catastrophic forgetting

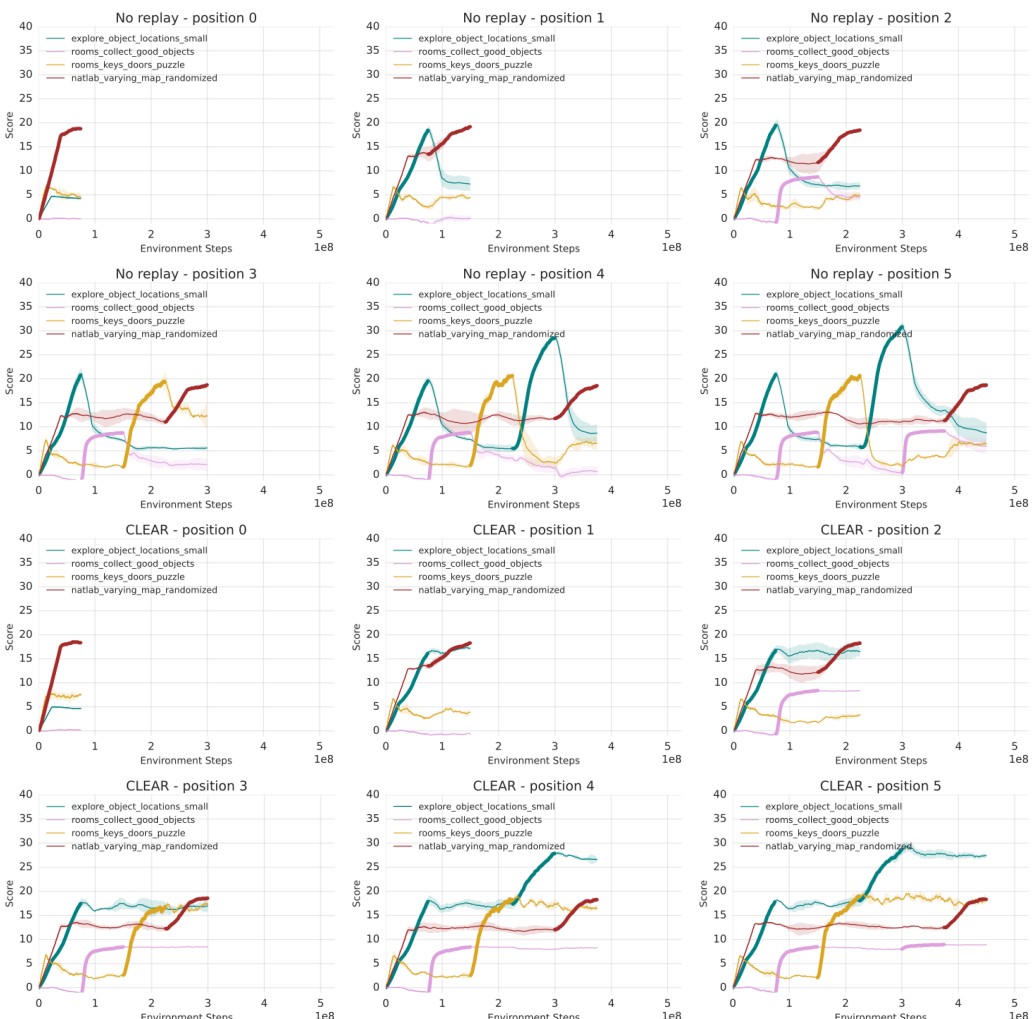

Figure 5: The DMLab task `natlab_varying_map_randomize` (brown line) is presented at different positions within a cyclically repeating sequence of three other DMLab tasks. We find that (1) performance on the probe task is independent of its position in the sequence, (2) the effectiveness of CLEAR does not degrade as more experiences and tasks are introduced into the buffer.

with no appreciable decrease in performance attained. Unless otherwise stated, our experiments on CLEAR will use a 50-50 split of new and replay data in training.

It is notable that it is possible to train purely on replay examples, since the network has essentially no on-policy learning. In fact, the figure shows that with 100% replay, performance on each task increases throughout, even when on-policy learning is being applied to a different task. Just as Figure 2 shows the importance of behavioral cloning for maintaining past performance on a task, so this experiment shows that off-policy learning can actually increase performance from replay alone. Both ingredients are necessary for the success of CLEAR.

## 4.4 LIMITED-SIZE BUFFERS

In some cases, it may be impractical to store all past experiences in the replay buffer. We therefore test the efficacy of buffers that have capacity for only a relatively small number of experiences (Figure 4). Once the buffer is full, we use reservoir sampling to decide when to replace elements of the buffer with new experiences (Isele & Cosgun, 2018) (see details in Appendix A). Thus, at each point in time, the buffer contains a (fixed size) sample uniformly at random of all past experiences.

We consider a sequence of tasks with 900 million environmental frames, comparing a large buffer of capacity 450 million to two small buffers of capacity 5 and 50 million. We find that all buffers perform well and conclude that it is possible to learn and reduce catastrophic forgetting even with a replay buffer that is significantly smaller than the total number of experiences. Decreasing the buffer size to 5 million results in a slight decrease in robustness to catastrophic forgetting. This may be due to over-fitting to the limited examples present in the buffer, on which the learner trains disproportionately often.

### 4.5 LEARNING A NEW TASK QUICKLY

It is a reasonable worry that relying on a replay buffer could cause new tasks to be learned more slowly as the new task data will make up a smaller and smaller portion of the replay buffer as the buffer gets larger. In this experiment (Figure 5), we find that this is not a problem for CLEAR, relying as it does on a mixture of off- and on-policy learning. Specifically, we find the performance attained on a task is largely independent of the amount of data stored in the buffer and on the identities of the preceding tasks. We consider a cyclically repeating sequence of three DMLab tasks. At different points in the sequence, we insert a fourth DMLab task as a "probe". We find that the performance attained on the probe task is independent of the point at which it is introduced within the training sequence. This is true both for normal training and for CLEAR. Notably, CLEAR succeeds in greatly reducing catastrophic forgetting for all tasks, and the effect on the probe task does not diminish as the probe task is introduced later on in the training sequence. See also Appendix B for an experiment demonstrating that pure off-policy learning performs quite differently in this setting.

### 4.6 COMPARISON TO P&C AND EWC

Finally, we compare our method to Progress & Compress (P&C) (Schwarz et al., 2018) and Elastic Weight Consolidation (EWC) (Kirkpatrick et al., 2017), state-of-the-art methods for reducing catastrophic forgetting that, unlike replay, assume that the boundaries between different tasks are known (Figure 6). We use exactly the same sequence of Atari tasks as the authors of P&C (Schwarz et al., 2018), with the same time spent on each task. Likewise, the network and hyperparameters we use are designed to match exactly those used in Schwarz et al. (2018). This is simplified by the authors of P&C also using a training paradigm based on that in Espeholt et al. (2018). In this case, we use CLEAR with a 75-25 balance of new-replay experience.

We find that we obtain comparable performance to P&C and better performance than EWC, despite CLEAR being significantly simpler and agnostic to the boundaries between tasks. On tasks `krull`, `hero`, and `ms_pacman`, we obtain significantly higher performance than P&C (as well as EWC), while on `beam_rider` and `star_gunner`, P&C obtains higher performance. It is worth noting that though the on-policy model (baseline) experiences significant catastrophic forgetting, it also rapidly re-acquires its previous performance after re-exposure to the task; this allows baseline to be cumulatively better than EWC on some tasks (as is noted in the original paper Kirkpatrick et al. (2017)). An alternative plot of this experiment, showing cumulative performance on each task, is presented in Appendix C.

## 5 DISCUSSION

Some version of replay is believed to be present in biological brains. We do not believe that our implementation is reflective of neurobiology, though there are potential connections; hippocampal replay has been proposed as a systems-level mechanism to reduce catastrophic forgetting and improve generalization as in the theory of complementary learning systems (McClelland, 1998). This contrasts to some degree with synapse-level consolidation, which is also believed to be present in biology (Benna & Fusi, 2016), but is more like continual learning methods that protect parameters.

Indeed, algorithms for continual learning may live on a *Pareto frontier*: different methods may have different regimes of applicability. In cases for which storing a large memory buffer is truly prohibitive, methods that protect inferred parameters, such as Progress & Compress, may be more suitable than replay methods. When task identities are available or boundaries between tasks are very clear, leveraging this information may reduce memory or computational demands or be useful to alert the agent to engage in rapid learning. Further, there exist training scenarios that are adversarial

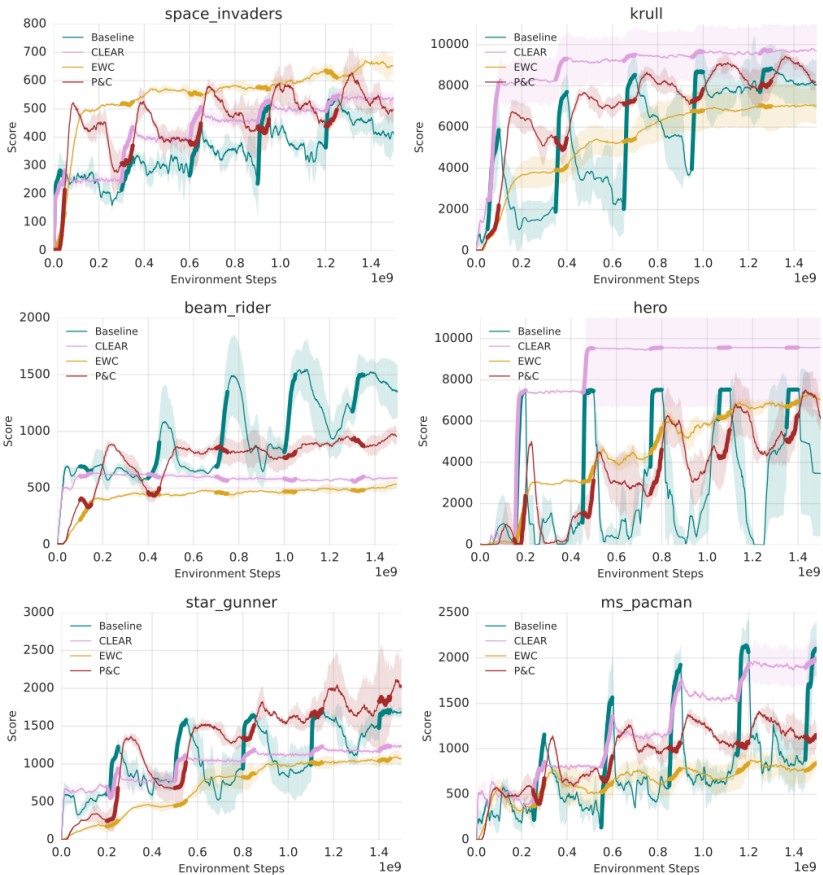

Figure 6: Comparison of CLEAR to Progress & Compress (P&C) and Elastic Weight Consolidation (EWC). We find that CLEAR demonstrates comparable or greater performance than these methods, despite being significantly simpler and not requiring any knowledge of boundaries between tasks. (See Appendix C for a different plot of these data.)

either to our method or to any method that prevents forgetting. For example, if the action space of a task were changed during training, fitting to the old policy's action distribution, whether through behavioral cloning, off-policy learning, weight protection, or any of a number of other strategies for preventing catastrophic forgetting, could have a deleterious effect on future performance. For such cases, we may need to develop algorithms that selectively protect skills as well as *forget* them.

We have explored CLEAR in a range of continual learning scenarios; we hope that some of the experimental protocols, such as probing with a novel task at varied positions in a training sequence, may inspire other research. Moving forward, we anticipate many algorithmic innovations that build on the ideas set forward here. For example, weight-consolidation techniques such as Progress & Compress are quite orthogonal to our approach and could be married with it for further performance gains. Moreover, while the V-Trace algorithm we use is effective at off-policy correction for small shifts between the present and past policy distributions, it is possible that off-policy approaches leveraging Q-functions, such as Retrace (Munos et al., 2016), may prove more powerful still.

We have described a simple but powerful approach for preventing catastrophic forgetting in continual learning settings. CLEAR uses on-policy learning on fresh experiences to adapt rapidly to new tasks, while using off-policy learning with behavioral cloning on replay experience to maintain and modestly enhance performance on past tasks. Behavioral cloning on replay data further enhances the agent's stability. Our method is simple, scalable, and practical; it takes advantage of the general abundance of memory and storage in modern computers and computing facilities. We believe that the broad applicability and simplicity of the approach make CLEAR a candidate "first line of defense" against catastrophic forgetting in many RL contexts.

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

## A   IMPLEMENTATION DETAILS

### A.1   DISTRIBUTED SETUP

Our training setup was based on that of Espeholt et al. (2018), with multiple actors and a single learner. The actors (which run on CPU) generate training examples, which are then sent to the learner. Weight updates made by the learner are propagated asynchronously to the actors. The workflows for each actor and for the learner are described below in more detail.

**Actor.** A training episode (*unroll*) is generated and inserted into the actor's buffer. Reservoir sampling is used (see further details below) if the buffer has reached its maximum capacity. The actor then samples another unroll from the buffer. The new unroll and replay unroll are both fed into a queue of examples that are read by the learner. The actor waits until its last example in the queue has been read before creating another.

**Learner.** Each element of a batch is a pair (new unroll, replay unroll) from the queue provided by actors. Thus, the number of new unrolls and the number of replay unrolls both equal the entire batch size. Depending on the buffer utilization hyperparameter (see Figure 3), the learner uses a balance of new and replay examples, taking either the new unroll or the replay unroll from each pair. Thus, no actor contributes more than a single example to the batch (reducing the variance of batches).

### A.2   NETWORK

For our DMLab experiments, we used the same network as in the DMLab experiments of Espeholt et al. (2018). We selected the shallower of the models considered there (a network based on Mnih et al. (2015)), omitting the additional LSTM module used for processing textual input since none of the tasks we considered included such input. For Atari, we used the same network in Progress & Compress (Schwarz et al., 2018) (which is also based on Espeholt et al. (2018)), also copying all hyperparameters.

### A.3   BUFFERS

Our replay buffer stores all information necessary for the V-Trace algorithm, namely the input presented by the environment, the output logits of the network, the value function output by the network, the action taken, and the reward obtained. Leveraging the distributed setup, the buffer is split among all actors equally, so that, for example, if the total buffer size were one million across a hundred actors, then each actor would have buffer capacity of ten thousand. All buffer sizes are measured in environment frames (not in numbers of unrolls), in keeping with the $x$-axis of our training plots.

For baseline experiments, no buffer was used, while all other parameters and the network remained constant.

Unless otherwise specified, the replay buffer was capped at half the number of environment frames on which the network is trained. This is by design – to show that even past the buffer capacity, replay continues to prevent catastrophic forgetting. When the buffer fills up, then new unrolls are added by reservoir sampling, so that the buffer at any given point contains a uniformly random sample of all unrolls up until the present time. Reservoir sampling is implemented as in Isele & Cosgun (2018) by having each unroll associated with a random number between 0 and 1. A threshold is initialized to 0 and rises with time so that the number of unrolls above the threshold is fixed at the capacity of the buffer. Each unroll is either stored or abandoned in its entirety; no unroll is partially stored, as this would preclude training.

## A.4 TRAINING

Training was conducted using V-Trace, with hyperparameters on DMLab/Atari tasks set as in Espeholt et al. (2018). Behavioral cloning loss functions $L_{\text{policy-cloning}}$ and $L_{\text{value-cloning}}$ were added in some experiments with weights of 0.01 and 0.005, respectively. The established loss functions $L_{\text{policy-gradient}}$, $L_{\text{value}}$, and $L_{\text{entropy}}$ were applied with weights of 1, 0.5, and $\approx 0.005$, in keeping with Espeholt et al. (2018). No significant effort was made to optimize fully the hyperparameters for CLEAR.

## A.5 EVALUATION

We evaluate each network during training on all tasks, not simply that task on which it is currently being trained. Evaluation is performed by pools of testing actors, with a separate pool for each task in question. Each pool of testing actors asynchronously updates its weights to match those of the learner, similarly to the standard (training) actors used in our distributed learning setup. The key differences are that each testing actor (i) has no replay buffer, (ii) does not feed examples to the learner for training, (iii) runs on its designated task regardless of whether this task is the one currently in use by training actors.

## A.6 EXPERIMENTS

In many of our experiments, we consider tasks that change after a specified number of learning episodes. The total number of episodes is monitored by the learner, and all actors switch between tasks simultaneously at the designated point, henceforward feeding examples to the learner based on experiences on the new task (as well as replay examples). Each experiment was run independently three times; figures plot the mean performance across runs, with error bars showing the standard deviation.

# B    PROBE TASK WITH 100% REPLAY

Our goal in this experiment was to investigate more thoroughly than Figure 3 to what extent the mixture of on- and off-policy learning is necessary, instead of pure off-policy learning, in learning new tasks swiftly. We rerun our "probe task" experiments (Section 4.5), where the DMLab task `natlab_varying_map_randomized` is presented at different positions in a cyclically repeating sequence of other DMLab tasks. In this case, however, we use CLEAR with 100% (off-policy) replay experience. We observe that, unlike in the original experiment (Figure 5), the performance obtained on the probe task `natlab_varying_map_randomized` deteriorates markedly as it appears later in the sequence of tasks. For later positions in the sequence, the probe task comprises a smaller percentage of replay experience, thereby impeding purely off-policy learning. This result underlines why CLEAR uses new experience, as well as replay, to allow rapid learning of new tasks.

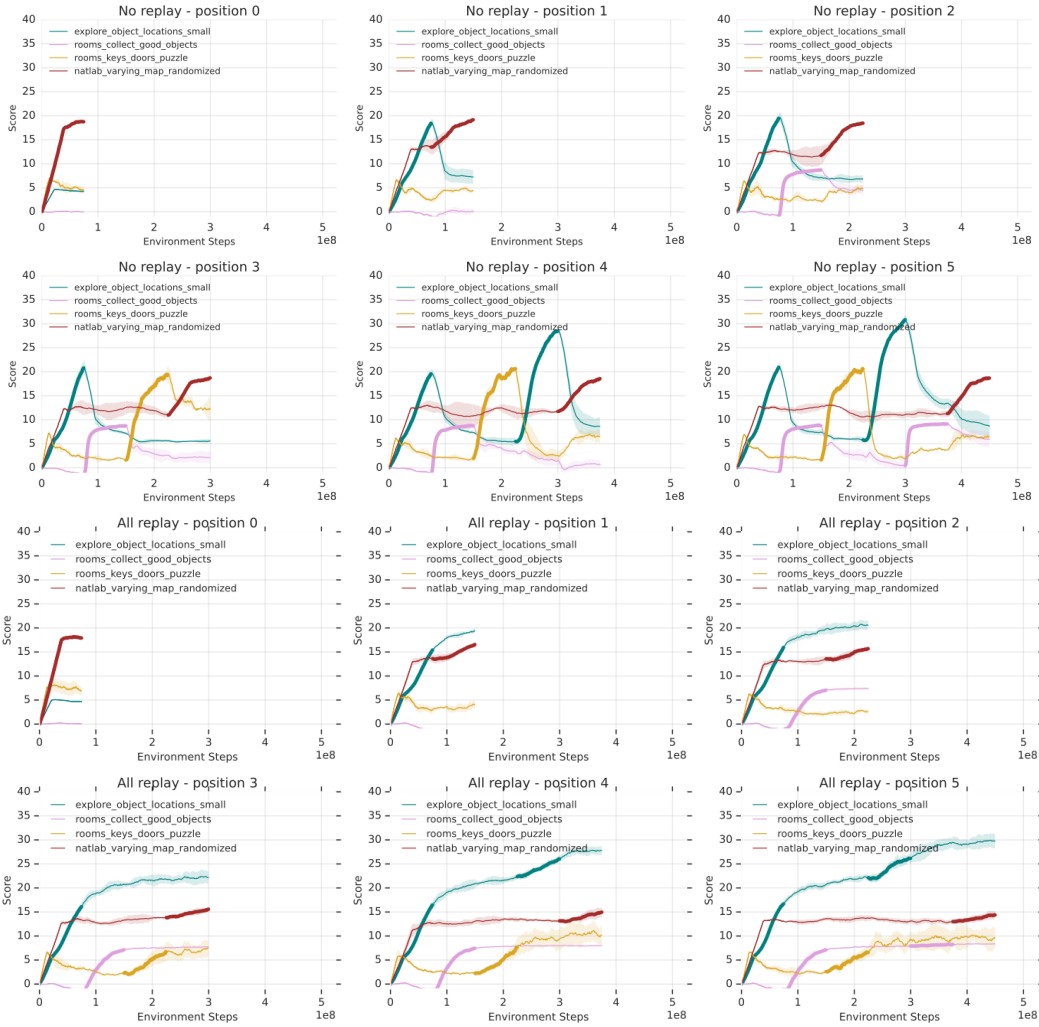

Figure 7: This figure shows the same "probe task" setup as Figure 5, but with CLEAR using 100% replay experience. Performance on the probe task `natlab_varying_map_randomized` decreases markedly as it appears later in the sequence of tasks, emphasizing the importance of using a blend of new experience and replay instead of 100% replay.

## C    FIGURES REPLOTTED ACCORDING TO CUMULATIVE SUM

In this section, we replot the results of our main experiments, so that the $y$-axis shows the mean cumulative reward obtained on each task during training; that is, the reward shown for time $t$ is the average $(1/t)\sum_{s<t} r_s$. This makes it easier to compare performance between models, though it smoothes out the individual periods of catastrophic forgetting. We also include tables comparing the values of the final cumulative rewards at the end of training.

|                              | explore... | rooms_collect... | rooms_keys... |
|------------------------------|------------|------------------|---------------|
| Separate                     | 29.24      | 8.79             | 19.91         |
| Simultaneous                 | 32.35      | 8.81             | 20.56         |
| Sequential (no CLEAR)        | 17.99      | 5.01             | 10.87         |
| CLEAR (50-50 new-replay)     | **31.40**  | **8.00**         | 18.13         |
| CLEAR w/o behavioral cloning | 28.66      | 7.79             | 16.63         |
| CLEAR, 75-25 new-replay      | 30.28      | 7.83             | 17.86         |
| CLEAR, 100% replay           | 31.09      | 7.48             | 13.39         |
| CLEAR, buffer 5M             | 30.33      | **8.00**         | 18.07         |
| CLEAR, buffer 50M            | 30.82      | 7.99             | **18.21**     |

Figure 8: Quantitative comparison of the final cumulative performance between standard training ("Sequential (no CLEAR)") and various versions of CLEAR (see Figures 9, 10, 11, and 12 below) on a cyclically repeating sequence of DMLab tasks. We also include the results of training on each individual task with a separate network ("Separate") and on all tasks simultaneously ("Simultaneous") instead of sequentially. As described in Section 4.1, these situations represent no-forgetting scenarios and thus present upper bounds on the performance expected in a continual learning setting, where tasks are presented sequentially. Remarkably, CLEAR achieves performance comparable to "Separate" and "Simultaneous", demonstrating that forgetting is virtually eliminated.

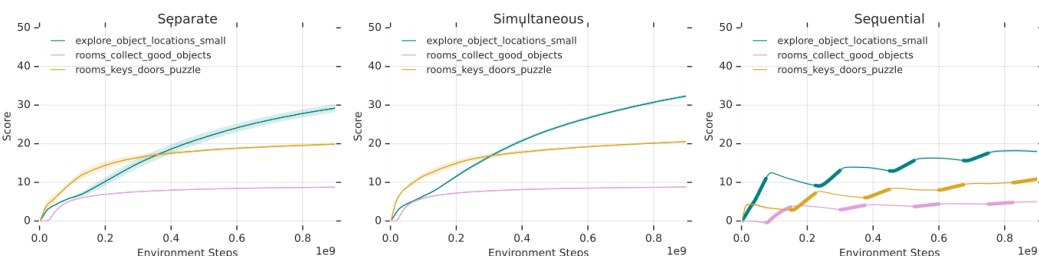

Figure 9: Alternative plot of the experiments shown in Figure 1, showing the difference in cumulative performance between training on tasks separately, simultaneously, and sequentially (without using CLEAR). The marked decrease in performance for sequential training is due to catastrophic forgetting. As in our earlier plots, thicker line segments are used to denote times at which the network is gaining new experience on a given task.

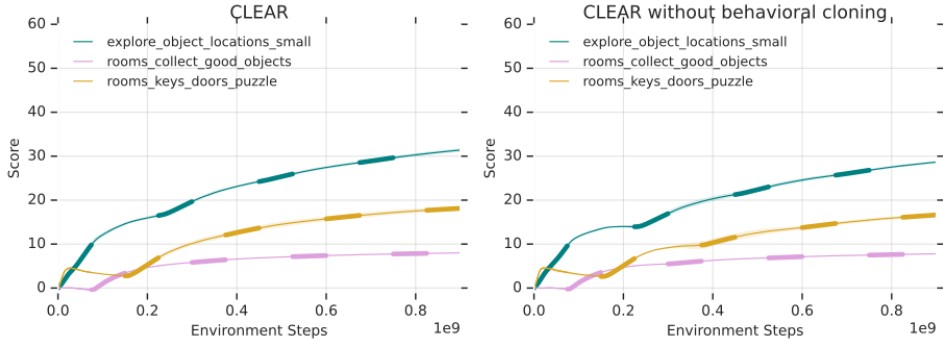

Figure 10: Alternative plot of the experiments shown in Figure 2, showing how applying CLEAR when training on sequentially presented tasks gives almost the same results as training on all tasks simultaneously (compare to sequential and simultaneous training in Figure 9 above). Applying CLEAR without behavioral cloning also yields decent results.

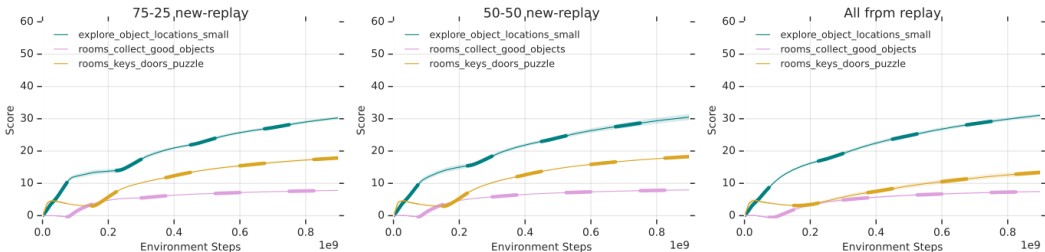

Figure 11: Alternative plot of the experiments shown in Figure 3, comparing performance between using CLEAR with 75-25 new-replay experience, 50-50 new-replay experience, and 100% replay experience. An equal balance of new and replay experience seems to represent a good tradeoff between stability and plasticity, while 100% replay reduces forgetting but lowers performance overall.

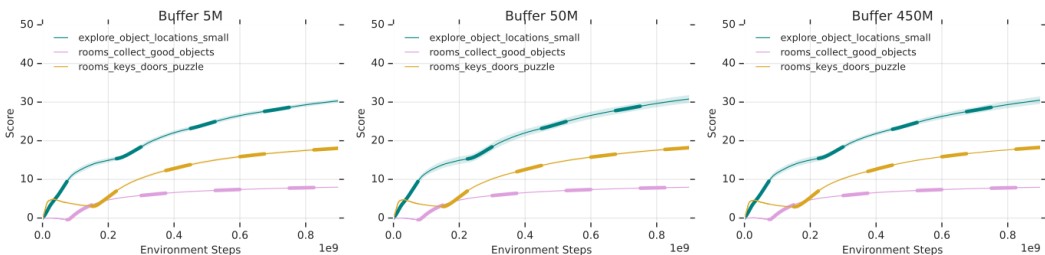

Figure 12: Alternative plot of the experiments shown in Figure 4, showing that reduced-size buffers still allow CLEAR to achieve essentially the same performance.

|  | space_invaders | krull | beam_rider | hero | star_gunner | ms_pacman |
|---|---|---|---|---|---|---|
| Baseline | 346.47 | 5512.36 | **952.72** | 2737.32 | 1065.20 | 753.83 |
| CLEAR | 426.72 | **8845.12** | 585.05 | **8106.22** | 991.74 | **1222.82** |
| EWC | **549.22** | 5352.92 | 432.78 | 4499.35 | 704.20 | 639.75 |
| P&C | 455.34 | 7025.40 | 743.38 | 3433.10 | **1261.86** | 924.52 |

Figure 13: Quantitative comparison of the final cumulative performance between baseline (standard training), CLEAR, Elastic Weight Consolidation (EWC), and Progress & Compress (P&C) (see Figure 14 below). Overall, CLEAR performs comparably to or better than P&C, and significantly better than EWC and baseline.

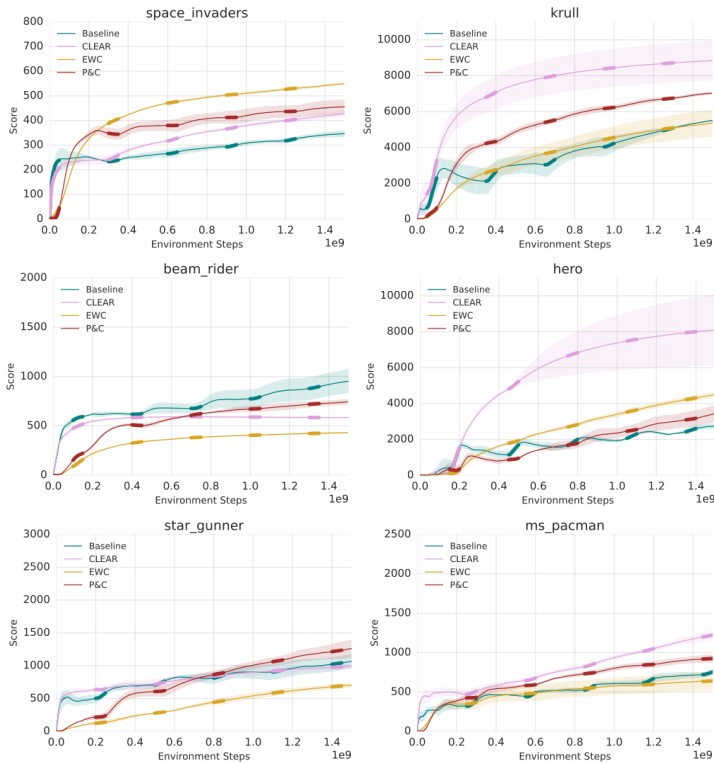

Figure 14: Alternative plot of the experiments shown in Figure 6, showing that CLEAR attains comparable or better performance than the more complicated methods Progress & Compress (P&C) and Elastic Weight Consolidation (EWC), which also require information about task boundaries, unlike CLEAR.

