# OpenReview forum: "Experience replay for continual learning"
_ICLR.cc/2019/Conference_

### Official Review · AnonReviewer3 · 2018-11-01
**Reduction in Catastrophic Forgetting by Augmented Experience Replay**

**Rating:** 5
**Confidence:** 5

**Review:**

The authors propose an approach to augment experience replay buffers with properties that can alleviate issues with catastrophic forgetting. The buffers are augmented by storing both new and historical experiences, along with the desired historical policy & value distribution. The AC learning now couples two additional losses that ensures the new policy does not drift away from old actor distribution (via KL) and new value does not drift away from old critic distribution (via L2 loss).

The authors provided clear experimental evidence that shows how an RL agent that does not use CLEAR will observe catastrophic when we sequentially train different tasks (and it is not due to destructive interference using the simultaneous and separate training/evaluation experiments). Author also showed how different replay make ups can change the result of CLEAR (and it's a matter of empirical tuning).

The formulation of CLEAR also is simple while delivering interesting results. It would have been nice to see how this is used in a practical setting as all these are synthetic environments / tasks. The discussion on relationship with biological mechanism also seems unnecessary as it's unclear whether the mechanism proposed is actually what's in the CLS.

---

> ### Author Response · Authors · 2018-11-27
> **Response to review, and notes on revision**
>
> We thank the reviewer for a thoughtful reading and for generally positive comments. As we understand, the reviewer's main concern is that the environments and tasks are synthetic. We would like to justify why we chose these particular environments and tasks. Our interest has been in reinforcement learning (RL) because, in the supervised learning case, catastrophic forgetting is generally resolved merely by storing a dataset. In RL, we view the synthetic problem setting as an absolutely mandatory first step to take before working on an application domain like robotics. Our experiments are in keeping with the standards in this field: work on continual learning to date in RL has all been on synthetic tasks, and we compare to this work, which allows us to provide effective benchmarks. The fact that our environments are simulated does not however imply that they are simple: they are state-of-the-art 3D environments, with DMLab introduced earlier this year and currently representing an advanced benchmark for RL systems.
>
> With respect to potential biological connections, we have devoted only a paragraph to this matter, with the purpose of emphasizing that we think such connections are NOT present, as the reviewer rightly states.
>
> In the revision, we have included in Appendix B the results of a new experiment, similar to that of the probe task (Figure 5).  In the new experiment, we show that when using pure off-policy learning (instead of a mixture of on- and off-policy learning, as in CLEAR), the probe task does indeed decrease in performance when other tasks are learned before it. CLEAR avoids this failure mode by blending new experience with replay.
>
> We have also, in our revision, added in Appendix C several new visualizations of our main results. In these figures, we plot the cumulative reward, which captures both performance and resistance to catastrophic forgetting, and we include tables that show the considerable benefit obtained by using CLEAR.

---

### Official Review · AnonReviewer1 · 2018-11-02

**Rating:** 5
**Confidence:** 4

**Review:**

The paper proposes a novel trial to alleviate the catastrophic forgetting for continual learning which is kind a mixture model of on and off-policy. The core concept of the method is utilizing experience replay buffer for all past events with new experience. They mainly worked on their method in the setting of reinforcement learning. In the experiments, they show that the model successfully mitigate the catastrophic forgetting with this behavioral cloning, and has the performance comparable to recent continual learning approaches.

The paper is easy to follow, and the methodology is quite intuitive and straight forward. In this paper, I have several questions.

Q1. I wonder the reason that every tasks are trained cyclically in sequence. And is there any trial to learn each task just once and observe the catastrophic forgetting of them when they have to detain the learned knowledge in a long time without training them again, as does most of visual domain experiments of the other continual learning research.

Q2. In figure 5, I wonder why the natlab_varying_map_ramdomize(probe task) can perform well even they didn’t learn yet. The score of brown line increases nearly 60~70% of final score(after trained) during training the first task. Because the tasks are deeply correlated? or it is just common property of probe task?

Q3. Using reservoir(buffer) to prevent catastrophic forgetting is natural and reasonable. Is there some of quantitative comparison in the sense of memory requirement and runtime? I feel that 5 or 50 million experiences at each task are huge enough to memorize and manage.

Additionally, in the experiment of figure 5, I think it could be much clear with a verification that the probe task  is semantically independent (no interference) over all the other tasks.

Also, it is quite hard to compare the performance of the models just with plots. I expect that it could be much better to show some of quantitative  results(as number).

---

> ### Author Response · Authors · 2018-11-27
> **Response to review, and notes on revision**
>
> We thank the reviewer for these comments and have made additional changes to the paper to address them, as we describe below.
>
> Q1: We presented tasks cyclically in sequence for several reasons. Presenting all tasks before returning to any one of them to represents a “worst-case” scenario for catastrophic forgetting and tests our method in the hardest situation.  Our experiment is designed to address exactly the scenario the reviewer describes -- spending a lot of time on the other tasks before returning to a specific one. The time spent on each task in the cycle is actually quite long, and if one imagines cutting off each figure after the first iteration of the cycle, one would end up with the figures the reviewer suggests. These figures would already provide ample support for all of our conclusions regarding CLEAR.
>
> Further, presenting tasks cyclically is a natural model of learning in which similar experiences are revisited over and over. Early researchers of human memory, e.g. Ebbinghaus, considered memorization tasks in which memorized items were recurrent and revisited several days in a row or over longer inter-experiment intervals. Recurrent study experiments permit the evaluation of several effects, including the phenomenon of savings, in which forgotten memories are rapidly re-acquired with marginal subsequent study. Here, we are also interested in demonstrating that repeated exposure to a task can be used to train the behavior of an agent.
>
> Q2: This is an interesting phenomenon! It is a demonstration of genuine constructive interference or positive transfer in which learning other tasks promotes coherent exploratory behavior in natlab_varying_map_randomize. This interference does not detract from the conclusions of the figure that catastrophic interference is present in this as in other tasks, that CLEAR fixes the problem, and that the ability of CLEAR to learn from new experience is unaffected by the amount of information already in the replay buffer.
>
> Q3: We understand the motivation behind this question, but the specific memory requirements depend on implementation, including the use of compression and caching techniques, which are engineering-level questions, and beyond the scope of what we can present in the paper, which is focused on the benefits that a mixture of on- and off-policy learning with behavioral cloning provides with respect to learning and forgetting. Notably, the buffer can almost certainly be compressed considerably given the commonalities between experiences. What memory requirements are unavoidable can leverage hard drive storage, with minimal RAM needed.
>
> Re Figure 5: The collection of results in the other figures are designed to show how a newly introduced task affects learning on other tasks. In this experiment, we were specifically interested in one question: Does having a full replay buffer from past experiences on other tasks slow learning on a new task? In Figure 5, note that the final performance obtained on the probe task doesn’t depend on whether it comes after another task, implying that learning the task is largely independent of the other tasks, except for the initial positive transfer.
>
> In the revision, we include a variation of the probe task experiment in Appendix B, in which we show that when using pure off-policy learning (instead of a mixture of on- and off-policy learning, as in CLEAR), the probe task does indeed decrease in performance when other tasks are learned before it. CLEAR avoids this failure mode by blending new experience with replay.
>
> Re numerical comparison: Absolutely, and we are very grateful for the suggestion. We have added tabulations of the cumulative sum of performance at the end of training for most experiments in Appendix C. We feel this measure captures both how quickly learning occurs and how much performance is maintained over the course of training on multiple tasks.

---

### Official Review · AnonReviewer2 · 2018-11-02
**Solid Paper, but Novelty over Experience Replay Needs Better Motivation**

**Rating:** 5
**Confidence:** 5

**Review:**

This paper proposes a particular variant of experience replay with behavior cloning as a method for continual learning. The approach achieves good performance while not requiring a task label. This paper makes the point that I definitely agree with that all of the approaches being considered should compare to experience replay and that in reality many of them rarely do better. However, I am not totally convinced when it comes to the value of the actual novel aspects of this paper.

Much of the empirical analysis of experience replay (i.e. the buffer size, the ratio of past and novel experiences, etc…) was not surprising or particular novel in my eyes. The idea of using behavior cloning is motivated fully through the lens of catastrophic forgetting and promoting stability and does not at all address achieving plasticity. This was interesting to me as the authors do mention the stability-plasticity dilemma, but a more theoretical analysis of why behavior cloning is somehow the right method among various choices to promote stability while not sacrificing or improving plasticity was definitely missing for me. Other options can certainly be considered as well if your aim is just to add stability to experience replay such a notion of weight importance for the past like in EwC (Kirkpatric et al., 2017) and many other papers or using knowledge distillation like LwF (Li and Hoeim, 2016). LwF in particular seems quite related. I wonder how LwF + experience replay compares to the approach proposed here. In general the discourse could become a lot strong in my eyes if it really considered various alternatives and explained why behavior cloning provides theoretical value.

Overall, behavior cloning seems to help a little bit based on the experiments provided, but this finding is very likely indicative of the particular problem setting and seemingly not really a game changer. In the paper, they explore settings with fairly prolonged periods of training in each RL domain one at a time. If the problem was to become more non-stationary with more frequent switching (i.e. more in line with the motivation of lifelong learning), I would imagine that increasing stability is not necessarily a good thing and may slow down future learning.

---

> ### Author Response · Authors · 2018-11-27
> **Response to review, and notes on revision**
>
> We thank the reviewer for a careful reading and for thoughtful comments. Our purposes in this work were to develop methods that are capable of mitigating catastrophic forgetting without using task identity or boundaries, while maintaining plasticity for learning from new experience. The reviewer has noted two excellent, related works, namely EWC and LwF, which we highlight in our literature review.  However, both of these approaches require information about task identities and boundaries. Furthermore, as we demonstrate in Figures 6 and 14, even with task boundaries, CLEAR performs considerably better than EWC. We also show similar performance to Progress & Compress, which itself represents a state-of-the-art improvement upon EWC (and also requires task boundaries, unlike our method). In our revision, we have provided in Appendix C a more detailed quantitative comparison of CLEAR against baseline methods.
>
> As the reviewer rightly points out, we are combining existing tools; the innovation is putting them together to make a highly effective, simple continual learning method. While in some respects, this application may be natural, it has evidently not been explored before, as evidenced by the fact that it outperforms state-of-the-art, highly-engineered systems.
>
> The reviewer rightly calls attention to the stability-plasticity dilemma. To restate the ingredients of CLEAR, we are combining: (1) learning from on-policy experience, which yields plasticity and adaptiveness on new tasks, (2) off-policy replay to learn from past experience, and (3) behavioral cloning to maintain past behavior.  Our experiments show that each of these ingredients is essential to our performance.
>
> 1 - On-policy learning. In a new experiment within the revision (Appendix B), we demonstrate that removing the on-policy learning component from our algorithm (so that only off-policy replay and behavioral cloning are used) significantly damages the plasticity of the method.  Our results show that when a new ‘probe’ task is introduced after learning several tasks, the probe task cannot be learned quickly without on-policy learning (note the difference between Figure 7 and Figure 5).
>
> 2 - Off-policy learning. In Figures 3 and 11, we show that CLEAR is able to learn reasonably well from pure replay experience, demonstrating the importance of the off-policy learning component.  (Clearly, behavioral cloning alone cannot increase performance on replay.)
>
> 3 - Behavioral cloning. Finally, in Figures 2 and 10, we show that leaving out behavioral cloning significantly damages the ability of CLEAR to maintain past performance.
>
> There is both literature on sequential task presentations meant to induce catastrophic forgetting and literature on replay. We bring these two worlds together, and show that the mixture of off- and on-policy learning, together with behavioral cloning, confers dramatic improvements for continual learning settings.

---

### Author Response · Authors · 2018-12-10
**Summary of changes made in revision**

We are grateful to all reviewers for their time and helpful suggestions. Our goal in this paper was to demonstrate that a surprisingly simple replay-based approach (CLEAR) dramatically reduces catastrophic forgetting in reinforcement learning tasks, improving upon existing methods while not requiring information about task boundaries.  The reviewers agree that the paper is "easy to follow" (R1) and that the method "achieves good performance" (R2) and is "simple while delivering interesting results" (R3).

R2 asks for clarification regarding the novelty of CLEAR over existing methods. We hope that the additional supplementary figures emphasize the improvement in performance offered by CLEAR, and in our response to R2 below we detail how each of the ingredients of CLEAR (on-policy learning, off-policy learning, and behavioral cloning) inform stability and/or plasticity. We also emphasize that replay has been thoroughly investigated in other contexts but (remarkably) has been largely ignored in the context of catastrophic forgetting, where we show that it is extremely powerful.

R1 asks for clarification on several questions regarding our experimental setup, and suggests tables for numerical comparison of CLEAR to other methods, which we have added in our revision. We have endeavored to answer the specific questions at length in our response to R1 below.

As we understand, R3's main concern is that the reinforcement learning experiments involve synthetic tasks, which we have motivated in our response to R3 below.

Changes made in our revision:

- We have included in Appendix B the results of a new experiment, similar to that of the probe task (Figure 5). In the new experiment, we show that when using pure off-policy learning (instead of a mixture of on- and off-policy learning, as in CLEAR), the probe task does indeed decrease in performance when other tasks are learned before it. CLEAR avoids this failure mode by blending new experience with replay.

- We have added in Appendix C several new visualizations of our main results. In these figures, we plot the cumulative reward, which captures both performance and resistance to catastrophic forgetting.

- We have also added in Appendix C tables that quantitatively summarize the considerable benefit obtained by using CLEAR.

---

### Meta-Review · Area_Chair1 · 2018-12-13
**Some insights into using ER to reduce catastrophic forgetting, but requires a bit better placement**

**Confidence:** 4
**Recommendation:** Reject

**Metareview:**

This paper and revisions have some interesting insights into using ER for catastrophic forgetting, and comparisons to other methods for reducing catastrophic forgetting. However, the paper is currently pitched as the first to notice that ER can be used for this purpose, whereas it was well explored in the cited paper "Selective Experience Replay for Lifelong Learning", 2018. For example, the abstract says "While various methods to counteract catastrophic forgetting have recently been proposed, we explore a straightforward, general, and seemingly overlooked solution – that of using experience replay buffers for all past events". It seems unnecessary to claim this as a main contribution in this work. Rather, the main contributions seem to be to include behavioural cloning, and do provide further empirical evidence that selective ER can be effective for catastrophic forgetting.

Further, to make the paper even stronger, it would be interesting to better understand even smaller replay buffers. A buffer size of 5 million is still quite large. What is a realistic size for continual learning? Hypothesizing how ER can be part of a real continual learning solution, which will likely have more than 3 tasks, is important to understand how to properly restrict the buffer size.

Finally, it is recommended to reconsider the strong stance on catastrophic interference and forgetting. Catastrophic interference has been considered for incremental training, where recent updates can interfere with estimates for older (or other values). This definition does not precisely match the provided definition in the paper. Further, it is true that forgetting has often been used explicitly for multiple tasks, trained in sequence; however, the issues are similar (new learning overriding older learning). These two definitions need not be so separate, and further it is not clear that the provided definitions are congruent with older literature on interference.

Overall, there is most definitely useful ideas and experiments in this paper, but it is as yet a bit preliminary. Improvements on placement, motivation and experimental choices would make this work much stronger, and provide needed clarity on the use of ER for forgetting.